# Effects of a Novel Infant Formula on Weight Gain, Body Composition, Safety and Tolerability to Infants: The INNOVA 2020 Study

**DOI:** 10.3390/nu15010147

**Published:** 2022-12-28

**Authors:** Julio Plaza-Diaz, Francisco Javier Ruiz-Ojeda, Javier Morales, Ana Isabel Cristina de la Torre, Antonio García-García, Carlos Nuñez de Prado, Cristóbal Coronel-Rodríguez, Cyntia Crespo, Eduardo Ortega, Esther Martín-Pérez, Fernando Ferreira, Gema García-Ron, Ignacio Galicia, María Teresa Santos-García-Cuéllar, Marcos Maroto, Paola Ruiz, Raquel Martín-Molina, Susana Viver-Gómez, Angel Gil

**Affiliations:** 1Department of Biochemistry and Molecular Biology II, School of Pharmacy, University of Granada, 18071 Granada, Spain; 2Instituto de Investigación Biosanitaria IBS.GRANADA, Complejo Hospitalario Universitario de Granada, 18014 Granada, Spain; 3Children’s Hospital of Eastern Ontario Research Institute, Ottawa, ON K1H 8L1, Canada; 4RG Adipocytes and Metabolism, Institute for Diabetes and Obesity, Helmholtz Diabetes Center at Helmholtz Center Munich, Neuherberg, 85764 Munich, Germany; 5Institute of Nutrition and Food Technology “José Mataix”, Centre of Biomedical Research, University of Granada, Avda. del Conocimiento s/n. Armilla, 18016 Granada, Spain; 6Product Development Department, Alter Farmacia SA, 28880 Madrid, Spain; 7CS Presentación Sabio, C/Alonso Cano 8, Móstoles, 28933 Madrid, Spain; 8Instituto Fundación Teófilo Hernando (IFTH), Parque Científico de Madrid, UAM. C/ Faraday 7, Edificio CLAID, 28049 Madrid, Spain; 9Departamento de Farmacología, Facultad de Medicina, Universidad Autónoma de Madrid, 28049 Madrid, Spain; 10Consulta Privada Carlos Núñez, C/Santiago Apóstol 10, Majadahonda, 28220 Madrid, Spain; 11Centro de Salud Amante Laffón, Distrito de Atención Primaria Sevilla, Servicio Andaluz de Salud, 41010 Sevilla, Spain; 12CAP Nova Lloreda, Av. De Catalunya 62-64, 08917 Badalona, Spain; 13CS Parque Loranca, C/de la Alegría 2, Fuenlabrada, 28942 Madrid, Spain; 14Consulta Externa Hospital Privado Santa Ángela de la Cruz, Av. De Jerez 59, 41013 Sevilla, Spain; 15CS La Rivota, C/de las Palmeras s/n, Alcorcón, 28922 Madrid, Spain; 16CS Las Américas, Av. De América 6, Parla, 28983 Madrid, Spain; 17CS Doctor Luengo Rodríguez, C/Nueva York 16, Móstoles, 28938 Madrid, Spain; 18CS Valle de la Oliva, C/Enrique Granados 2, Majadahonda, 28222 Madrid, Spain; 19CIBEROBN (CIBER Physiopathology of Obesity and Nutrition), Instituto de Salud Carlos III, 28029 Madrid, Spain

**Keywords:** arachidonic acid, α-lactalbumin, *Bifidobacterium animalis* subsp. *lactis*, BPL1^TM^, body composition, docosahexaenoic acid, infant formula, postbiotics, protein, safety

## Abstract

Exclusive breastfeeding is recommended for the first six months of life to promote adequate infant growth and development, and to reduce infant morbidity and mortality. However, whenever some mothers are not able to breastfeed their infants, infant formulas mimicking human milk are needed, and the safety and efficacy of each formula should be tested. Here, we report the results of a multicenter, randomized, blinded, controlled clinical trial that aimed to evaluate a novel starting formula on weight gain and body composition of infants up to 6 and 12 months, as well as safety and tolerability. For the intervention period, infants were divided into three groups: group 1 received formula 1 (Nutribén^®^ Innova 1 (Alter Farmacia S.A., Madrid, Spain) or INN (*n* = 70)), with a lower amount of protein, a lower casein to whey protein ratio by increasing the content of α-lactalbumin, and a double amount of docosahexaenoic acid/arachidonic acid than the standard formula; it also contained a thermally inactivated postbiotic (*Bifidobacterium animalis* subsp. *lactis*, BPL1^TM^ HT). Group 2 received the standard formula or formula 2 (Nutriben^®^ Natal (Alter Farmacia S.A., Madrid, Spain) or STD (*n* = 70)) and the third group was exclusively breastfed for exploratory analysis and used as a reference (BFD group (*n* = 70)). During the study, visits were made at 21 days and 2, 4, 6, and 12 months of age. Weight gain was higher in both formula groups than in the BFD group at 6 and 12 months, whereas no differences were found between STD and INN groups either at 6 or at 12 months. Likewise, body mass index was higher in infants fed the two formulas compared with the BFD group. Regarding body composition, length, head circumference and tricipital/subscapular skinfolds were alike between groups. The INN formula was considered safe as weight gain and body composition were within the normal limits, according to WHO standards. The BFD group exhibited more liquid consistency in the stools compared to both formula groups. All groups showed similar digestive tolerance and infant behavior. However, a higher frequency of gastrointestinal symptoms was reported by the STD formula group (*n* = 291), followed by the INN formula (*n* = 282), and the BFD groups (*n* = 227). There were fewer respiratory, thoracic, and mediastinal disorders among BFD children. Additionally, infants receiving the INN formula experienced significantly fewer general disorders and disturbances than those receiving the STD formula. Indeed, atopic dermatitis, bronchitis, and bronchiolitis were significantly more prevalent among infants who were fed the STD formula compared to those fed the INN formula or breastfed. To evaluate whether there were significant differences between formula treatments, beyond growth parameters, it would seem necessary to examine more precise health biomarkers and to carry out long-term longitudinal studies.

## 1. Introduction

Exclusive breastfeeding is the gold standard for infant feeding because it promotes adequate growth and development, excellent nutritional status, and appropriate psychological development. In addition, due to the special composition of breast milk in bioactive and immunogenic substances, it effectively protects against numerous infectious diseases, mainly pneumonia and other respiratory infections, diarrhea, and allergic processes. Moreover, breastfeeding promotes an optimal psycho-affective mother–child relationship and has a very low cost, practically nil [1,2,3,4,5]. Indeed, WHO and UNICEF recommend starting breastfeeding within the first hour of birth and being exclusively breastfed for the first 6 months of life [6,7]. However, from the age of 6 months, children should begin eating safe and adequate complementary foods while continuing to breastfeed for up to 2 years and beyond [8].

Breastfeeding protects against disease in both developing [1] and developed countries [2] and is beneficial not only for infants but also for mothers. Indeed, breastfeeding may protect later in life against obesity and metabolic diseases [7,9]. Additionally, breastfeeding is associated with better performance on intelligence tests [10,11]. Furthermore, women who breastfeed have a reduced risk of breast and ovarian cancers [7].

Despite worldwide efforts to promote breastfeeding, there are many social factors, especially premature return to work after childbirth and, in some cases, maternal illness, that cause many mothers to abandon breastfeeding prematurely [12]. To promote the adequate growth and development of general infants and those who have those conditions, infant formula designed with an optimal nutritional composition is essential. In this regard, different entities such as the European Society for Paediatric Gastroenterology, Hepatology, and Nutrition (ESPGHAN) [13] and the American Academy of Paediatrics (AAP) have established various recommendations for the composition of infant formulas. In addition, various governmental organizations, such as the FAO/WHO Codex Alimentarius Commission [14], the European Union Commission (Commission Delegated Regulation (EU) 2016), and the US Food and Drug Administration (FDA) [15] have established strict compositional and control standards for infant formula foods to protect the health of consumers and ensure fair practices in the food trade.

In human milk, 87% of the mass consists of water, 1% protein, 4% lipids, and 7% carbohydrates (of which 1 to 2.4% are oligosaccharides). A variety of minerals are also present in it (calcium, phosphorus, magnesium, potassium, sodium, chloride, and several trace elements) as well as all necessary vitamins. Some minor proteins are more prevalent in human milk (lysozyme, lactoferrin, etc.) as well as the nonprotein nitrogen fraction (urea and free amino acids, including taurine, nucleotides, etc.). As a result, the protein content of human milk is low (10 g/L) [16].

Continuous research on the composition of human milk and the biological effects of its components [17] has led to the constant evolution of infant formula, especially during the last five decades, incorporating various food ingredients not only to meet the nutritional needs of infants but also to contribute to better development and functionality [18]. Thus, based on certain studies suggesting that a high protein intake in the early stages of life may be the cause of obesity and increased risk of metabolic disease in later stages of life [19,20], the protein composition of infant formulas has been adjusted in both quality and quantity, reducing the protein intake of infants [21,22,23,24]. Regarding the quality of protein intake, the whey/casein ratio in infant formulas is important for the first year of life. Whey proteins, both β-lactoglobulin and α-lactalbumin, are rapidly digested and participate in the building of muscle mass [25], with α -lactalbumin being the major protein in human milk [26]. Therefore, the development of infant formula containing bovine α-lactalbumin may improve the plasma amino acid pattern of the receiver infant, allowing a reduction in the protein content of the formula. On the other hand, the relatively high content of long-chain polyunsaturated fatty acids of both the *n*-6 and *n*-3 series, especially arachidonic acid (AA, 20:4 *n*-6) and docosahexaenoic acid (DHA, 22:6 *n*-3) in human milk and their proven effects on the cognitive development of infants, has led to the incorporation of these fatty acids into infant formulas. In this regard, and even though EFSA and the European Union Commission have established mandatory compositional recommendations only for DHA content [27,28], numerous researchers have argued and established consensus documents on the need for infant formulas to incorporate both AA and DHA in amounts comparable to the average composition of human milk. Hence, according to some studies infant formula should provide DHA at levels of 0.3% to 0.5% by weight of total fatty acids and with a minimum level of ARA equivalent to the DHA content [29,30]. Therefore, the supplementation of infant formulas with both DHA and ARA in appropriate amounts should be clinically tested.

During the last two decades, advances in the knowledge of the composition of the intestinal microbiome of breastfed infants and its beneficial effects on the host [31,32] have led to the incorporation of both prebiotics and probiotics in infant formulas [33,34]. In this regard, improvements in the prevention of acute infectious diarrhea, treatment of antibiotic-associated diarrhea, and prevention of allergy have been reported [33,34,35,36]. Also, it is known that not only live probiotics but also their fermentation products and dead cells derived from inactivated probiotics, referred to by the International Scientific Association of Probiotics and Prebiotics (ISAPP) as postbiotics, can exert biological effects of interest on the intestinal microbiota [37,38]. In this context, *Bifidobacterium animalis* subsp. *lactis* CECT 8145 inactivated with heat (BPL1^TM^ HT) exhibits several biological effects of interest such as reduction in fat deposition via the IGF-1 pathway in mice [39] and has anti-obesity effects in obese Zucker rats [40]. Moreover, daily consumption of the living strain BPL1^TM^ ameliorates several anthropometric adiposity biomarkers in abdominally obese adults [41]. Likewise, an infant formula supplemented with BPL1^TM^ HT reduced fat deposition in *C. elegans* and augmented acetate and lactate in a fermented infant slurry [42].

Novel infant formulas should be thoroughly evaluated to ensure their safety, efficacy, and tolerability. Hence, in the present study, we aimed to evaluate the effect of a novel starting infant formula on weight gain, body composition, safety, and tolerability, in infants up to 6 and 12 months of age. This was compared with standard infant formula. Safety and tolerability outcomes included digestive tolerance (flatulence, vomiting, and regurgitation), stool appearance (consistency and frequency), behavior (restlessness, colic, and nocturnal awakenings), and incidence of infections. Safety objectives included any frequency of gastrointestinal symptoms resulting from the consumption of the study formula. As a further exploratory objective, we compared weight gain up to 6 and 12 months and changes in the other secondary outcomes among the infants who received the modified starter formula and the standard formula with a group of exclusively breastfed infants. The complete design and methodology of this randomized, multicenter, double-blind, parallel, comparative clinical trial was previously registered with Clinicaltrial.gov (NCT05303077) on 31 March 2022, last updated on 7 April 2022, and published [43].

## 2. Materials and Methods

### 2.1. Ethics

This clinical trial was carried out following the recommendations of the International Conference on Harmonization Tripartite on PCBs, the ethical–legal principles established in the latest revision of the Declaration of Helsinki, as well as the current regional regulations that regulate pharmacovigilance and food safety. The present study was approved by the Committee for Technical Investigation in Regional Medicine in the Madrid Community (CEIm-R) dated 11 May 2018 under the name INNOVA2020 version 2.0. All personal data obtained in this study are confidential. They were treated under the Spanish Organic Law 3/2018, of December 5, on the Protection of Personal Data and guarantee of digital rights. The researchers or the institutions implicated in the study allowed direct access to the data or source documents for monitoring, auditing, and review by the CEIm-R. They also allowed inspection of the trial by health authorities.

### 2.2. Trial Design

The INNOVA study was designed as a randomized, multicenter, double-blind, parallel, and comparative clinical trial of the equivalence of two starting formulas for infants. Furthermore, a third un-blinded group of breastfed infants was used as a further reference group for exploratory analysis. Blinding for both investigator and participant remained assured as both infant formulas were labeled the same. It is not mandatory to carry out specific clinical tests to demonstrate the nutritional and healthy properties of infant formulas as per the current EU legislation (EC Regulation No. 1924/2006) [43].

Parents were informed by pediatricians involved in the study of the possibility of participating in it at a meeting at 15 days of infant age. After agreeing to participate, they came to the health center for the baseline visit a week later (visit at 21 days of age). The pediatrician requested, at the meeting at 15 days of life, that parents provide at the next visit relevant information from the maternal history that was required for the study and that was not available in the pediatric history. The visit that took place within the scheduled time was considered valid with a margin of ±3 days for the visit at 21 days of age, ±1 week for the visit at 2 months of age, and ±2 weeks for visits at 4, 6, and 12 months of age. The inclusion criteria were healthy children of both sexes; term children (between 37 and 42 weeks of gestation), birth weight between 2500 and 4500 g, single delivery, and mothers with a body mass index, before pregnancy, between 19 and 30 kg/m^2^. The exclusion criteria were body weight less than the 5th percentile for that gestational age at the time of inclusion, allergy to cow’s milk proteins and/or lactose, history of antibiotic use during the 7 days before inclusion, congenital disease or malformation that can affect growth, diagnosis of disease or metabolic disorders, significant prenatal and/or severe postnatal disease before enrollment, minor parents (younger than 18 years old), newborn of a diabetic mother, newborn of a mother with drug dependence during pregnancy, newborn whose parents/caregivers cannot comply with procedures of the study, and infants participating or having participated in another clinical trial since their birth. Participants had the right to withdraw from the study at any time and for any reason, without giving explanations or suffering any penalty for it. Likewise, the investigator could withdraw study participants if they did not comply with the study procedures. This was for any reason that, in the investigator’s opinion, may have posed a risk to the infant or made it necessary to suspend breastfeeding. Children who received antibiotics during the 7 days before inclusion were not considered eligible for the study and children who received antibiotics during the trial were excluded. Infants were also excluded if they received any treatment, food, or product that could interfere with the trial at any time or breastfeeding that was different from that of the group assigned in the trial.

The sample size of the trial was 210 children (70/group), based on the main outcome of weight gain, which was the main variable chosen according to the “Guidelines from the American Academy of Pediatrics Task Force on Clinical Testing of Infant Formulas” [44]. The infants were selected by primary care pediatricians through active and consecutive recruitment; i.e., pediatricians informed and invited parents of 15-day-old infants not being breastfed for some reason to be involved in the trial (see details below under item 2.3). The study was carried out in 21 centers, all located in Spain, of which 17 recruited at least one subject. In total, 217 subjects signed the informed consent (IC) and 145 were randomized to receive one of the two infant formulas. Of these 145, 3 were randomization failures and 2 were screening failures; thus 140 infants who met all the inclusion criteria and no exclusion criteria were included in the study, and 70 were unblinded within the infants exclusively breastfed group. A total of 185 subjects completed all study visits, of which there were 25 dropouts, 12 in the breastfeeding group, 8 in the formula 1 group, and 5 in the standard formula group. The trial was registered with the clinicaltrial.gov website (NCT05303077, https://clinicaltrials.gov/ct2/show/NCT05303077, accessed on 1 July 2022) on 31 March 2022 and last updated on 7 April 2022 [43]. The inclusion period started on 01 October 2018 and the follow-up period was between 11 February 2019 and 25 November 2020.

### 2.3. Formula Characteristics

Group 1 (Infant formula 1): Nutribén^®^ Innova 1 (Alter Farmacia S.A., Madrid, Spain)Group 2 (Infant formula 2): Nutribén^®^ Standard (Alter Farmacia S.A., Madrid, Spain)Group 3: Breastfeeding (External control exploratory analysis)

Infants were recruited from primary care pediatric clinics by the pediatricians participating in the trial. Pediatricians informed and invited parents of 15-day-old infants who visited their offices regularly (for regular medical check-ups) to be involved in the trial. Infants that were not receiving breastfeeding at the time of inclusion (for different reasons) were proposed to participate in the formula-feeding groups. To keep the three arms of the trial balanced, one candidate breastfeeding subject was recruited at each center for every two infants supplemented with infant formula. The selection of the children was made among those infants who met the inclusion and exclusion criteria of the study and fit the categories of sex, body mass index (BMI) of the mother before pregnancy (<25 kg/m^2^ or >25 kg/m^2^), and birth weight (<3500 g or >3500 g).

The experimental product object of this trial (Formula 1 or Nutribén^®^ Innova 1, (Alter Farmacia S.A., Madrid, Spain, INN formula) and the Nutribén^®^ Standard formula (Alter Farmacia S.A., Madrid, Spain, STD formula) comply with the recommendations of the ESPGHAN (European Society of Pediatric Gastroenterology, Hepatology, and Nutrition) and with Regulation 609/2013 of the European Parliament and of the Council regarding foods intended for children, infants, and young children, foods for special medical purposes and complete diet substitutes for weight control and repealing Council Directives 92/51, Directives 96/8/EC, 1999/21/CE, 2006/125/CE, and 2006/141/CE of the Commission, Directive 2009/38/CE of the European Parliament and of the Council, and Regulations 41/2009 and 953/2009 of the Commission. More detailed information on the composition of each of the products can be found in Table 1. Infant formulas were given ad libitum orally. The two trial formulations were administered following the preparation instructions in the manufacturer’s package insert. DHA was obtained from purified and concentrated fish oil.

### 2.4. Measurements and Evaluations

#### 2.4.1. Anthropometric Measures

Weight, length, head circumference, body mass index, body fat percentage, and lean body mass measurements were taken at all study visits (visit 1, 21 days, visit 2, 2 months, visit 3, 4 months, visit 4, 6 months, and visit 5, 12 months). Fat mass was estimated using skinfold measurements. From the visit scheduled for the fourth month of life, the mean arm circumference, the triceps skinfold, and the subscapular skinfold were added to the previous measurements. All these measurements were done in duplicate using as valid data the mean between the two. Each center had all the necessary materials to carry out these measurements. They needed to always use the same material with all subjects and the material needed to be calibrated.

#### 2.4.2. Characteristics of Feces and Digestive Tolerance

We evaluated consistency and the number of stools/day at 21 days and 2, 4, 6, and 12 months of age. Flatulence, vomiting, and regurgitation at 21 days and 2, 4, 6, and 12 months of age were also assessed.

#### 2.4.3. Infants’ Behavior Evaluated by Parents

Restlessness, colic, and night awakenings were evaluated during 21 days and 2, 4, 6, and 12 months of age for the parents. More details are available in the study protocol [43].

#### 2.4.4. Morbidity, Safety, and Tolerability

Variables related to morbidities were evaluated within 1 year. All morbidities were coded according to the MedDRA dictionary version 21.0 [45].

### 2.5. Statistical Methods

The main objective of the study was to evaluate whether the weight gain was equivalent between treatment groups receiving formula 1 and 2, and it was decided that the sample size of the trial would be 210 children (70/group) based on the primary variable weight gain, following the “Guidelines from the American Academy of Pediatrics” issued by the Task Force on Clinical Testing of Infant Formulas of the American Academy of Pediatrics [44]. Previous studies carried out on children fed from 0 to 6 months with different formulations of infant formula have shown a mean weight gain of around 20–25 g/day with a standard deviation between 5 and 6 g/day [46]. In most of these studies, a difference in mean weight gain of 3 g/day was considered clinically relevant. Thus, the main objective was resolved using a t-test for independent samples. Considering a power of 80%, a significance level of 5%, an equivalence limit of 3 g/day, and a common standard deviation of 5.5 g/day, we needed to recruit at least 59 children in each of the groups. Based on an estimated dropout rate of 20%, it was necessary to include 70 children in each of the groups, which means 140 infants. A third group of the same size (70 children) was included for the secondary comparisons between the bottle-fed and breastfed groups, maintaining the same significance and power in this secondary comparison as in the main comparison. Categorical variables are described as frequencies or percentages. Continuous variables are reported as mean and standard deviation. Between-group differences in measures of growth and body composition were analyzed using analysis of variance, including analysis of covariance and general models of analysis of variance for repeated measures (GLM-ANOVA and MANOVA) when required. The Chi-square test was applied to compare discrete variables between groups. The Bonferroni correction was used when comparisons were made between more than two groups. The alpha level of significance was set at 0.05. All evaluable infants were included in the statistical analysis, considering “evaluable” all those who had the main measurement variable. Both intention-to-treat and protocol-based statistical analyses were performed.

## 3. Results

Table 2 shows demographic and descriptive information for infants and their mothers. Here, gestational age, birth weight, childbirth delivery, assisted reproductive technology and the mother’s medical background were included, though no significant variables were shown. During visit 1 (21 days), only daily depositions per day were significantly higher for the BFD group in comparison with formula groups.

The parents or legal guardians returned to the center all the leftover containers (empty containers and containers with products) to record the amount of product dispensed and the amount of infant formula used up to the time of the visit; the corresponding weighing was performed and the percentage of product ingested by the participants was determined. Finally, the grams of formula consumed by each infant, measured by weighing the leftover formula or as indicated in the parents’ diary corresponding to the periods between visits 1–2, 2–3, and 3–4, were available. The participants of this study included in the formula-feeding groups were considered compliant if they took at least 80% of the formula during the first six months while in the breastfeeding group it was considered a valid external control if more than 80% of the feedings were breastfeeding. All infants included in the BFD group, except one, consumed more than 80% human milk during the first six months.

Based on the record of the amount of product dispensed and the amount of infant formula used (measurement of leftover) we estimated the average daily intake from visits 1 to 4 for the STD group (130.2 ± 14.7 g/d, equivalent to 671 ± 76 kcal/d of energy and 13.8 ± 1.56 g/d of protein) and the INN group (134.0 ± 18.2 g/d, equivalent to 655 ± 89 kcal /d of energy and 12.60 g/d of protein).

A GLM-ANOVA was performed for grams of formula consumed per day, and significant differences were observed for visit, sex, and infant birth weight, but not the formula used (data not presented).

The primary variable (weight gain) consisted of the difference in infant weight (in g/day) between the initial recruitment visit and the 6-month visit, divided by the number of days between the two visits. This analysis was performed on the 187 infants who completed until visit 4 (6 months). The difference between visit 1 and visit 4 was calculated in days of 187 children individually (Table 3). When analyzing the mean difference in weight gain between the groups of children treated with the STD and INN formulas (Table 3), there was no statistically significant difference in weight gain between the two formulations. On the other hand, an ANOVA of the weight difference between visits 1 and 4 was performed for the STD and INN formulas controlling for gender, maternal body mass index (BMI), and birth weight. There were significant differences in weight difference between visits 1 and 4 as a function of gender and infant birth weight, but not as a function of formula or mother’s BMI (Table 3). The grams consumed per day were also related to the difference in weight between visits 1 and 4. Based on the weighing of leftover dispensed formula, a multivariate analysis of variance (MANOVA) of the difference in weight of the children between visits 1 and 4 was performed as a function of formula consumed (STD, INN), gender (male, female), maternal BMI (<25, >25), birth weight (<3500, >3500), and grams ingested per day. A moderate positive correlation (*r* = 0.535) was observed between mean grams ingested per day (measured by weighing leftover formula) and weight gain between visits 1 and 4 of the infants. This correlation changed significantly by gender (males achieved larger weight differences for the same daily grams of formula consumed) and by birth weight (infants with birth weights > 3500 g achieved larger weight differences for the same daily grams of formula consumed) but not as a function of maternal BMI or formula used.

This same MANOVA as a function of grams ingested daily according to the parental diary also showed a positive correlation (*r* = 0.626) between mean grams ingested per day (obtained from parental diary entries) and weight gain between visits 1 and 4 of the infants. This correlation changed significantly by gender (boys achieved larger weight differences for the same daily grams of formulation consumed) and by birth weight (children with birth weights >3500 achieved larger weight differences, especially those who consumed more daily grams of formula) but not as a function of maternal BMI or formula used.

Secondarily and as foreseen in the study protocol, the weight gain of the infants who received each of the formulas studied was compared with the BFD group. The results of this analysis are shown in Table 3, where it can be seen that BFD infants had a significantly lower weight gain at 6 months of age than that shown by infants fed with either of the two formulas evaluated (STD and INN).

The same analysis was performed for the variable weight gain at 12 months of study. Table 3 shows the number of children in each of the three groups studied with weight values recorded at visit 1 (21 days of age) and visit 5 (12 months ± 2 weeks). A comparison of the means of weight gain from the different formulations (STD vs. INN, STD vs. BFD, and INN vs. BFD, Table 3) revealed that at 6 months there was no significant difference in weight gain (measured in g/day) as well as at 12 months between the STD and INN formulas. The total weight of the STD formula group remained significantly higher than the BFD group at 12 months; however, the INN group exhibited a trend to be different from breastfeeding between visits 1 and 5 (*p* = 0.057). A GLM-ANOVA (data not presented) showed significant differences for visit (*p* < 0.001) and for the interaction between formula and visit (*p* = 0.004), but not for formula exclusively (*p* = 0.381). The largest effect size was associated with the visit factor.

No significant differences were observed in length and head circumference, as well as tricipital and subscapular skinfolds and mean upper arm circumference (Table 4 and Table 5).

In the case of BMI, the ANOVA (Table 6) showed significant differences for visit (*p* < 0.001) and for the interaction between formula and visit (*p* < 0.001), but not for formula (*p* = 0.487). The largest effect size was associated with the visit factor (0.4230, data not provided). In addition, we have reported that it was from the third visit onwards that the mean BMI was higher in the STD and INN formulas compared to the BFD group; hence, the interaction effect appeared as significant (Table 6). For the body fat percentage, the ANOVA (Table 6) showed significant differences for visit (*p* = 0.005), but not for the interaction between the formula and visit (*p* = 0.958) or for formula (*p* = 0.249). The largest effect size was associated with the visit factor (0.0236, data not shown). Finally, for the lean mass, the ANOVA (Table 6) showed significant differences for visit (*p* < 0.001), but not for the interaction between formula and visit (*p* = 0.054) or for formula (*p* = 0.215). The largest effect size was associated with the visit factor (0.6093, data not provided).

In the case of BMI percentiles, the ANOVA (Appendix A) revealed significant differences for visit (*p* = 0.009) and the interaction between formula and visit (*p* = 0.006), but not for formula (*p* = 0.504). The largest effect size was associated with the visit factor (0.0222, data not provided). In the ANOVA, there were significant differences in the height percentiles for visit (*p* = 0.011), for the interaction between formula and visit (*p* < 0.001), but not for formula (*p* = 0.433, Appendix A). The largest effect size as associated with the visit factor (0.0154, data not shown). Finally, in the weight percentiles, the ANOVA (Appendix A) showed significant differences for visit (*p* < 0.001) and for the interaction between formula and visit (*p* < 0.001), but not for formula (*p* = 0.368). The largest effect size was associated with the visit factor (0.0228, data not provided).

The time course of the study showed a difference between breastfed infants presenting stools of a more liquid consistency (Appendix A), although these differences ceased to be significant at visit 5 (see Appendix A). Likewise, the daily number of stools was higher in breastfed infants at visits 1, 2, and 3, but these differences ceased to be significant at visits 4 and 5 (Table 7).

Figure 1 shows the tabulation of the different categories of digestive tolerance between every two visits. An increase in the percentage of children with high tolerance was seen throughout the study without significant differences between the different formulas and compared with the BFD group.

The infant’s behavior was re-coded as altered mood or pleasant mood according to the guardians or parents. Appendix A shows how the percentage of infants with altered behavior decreased throughout the study. No significant differences were observed among the three ways of feeding.

The study found no differences in tolerability between the different groups (Appendix A). Regarding to the safety of the different formula, the majority of the adverse events (*n* = 754) were mild—Grade 1 (94.3%), 13 were moderate—Grade 2 (1.6% of the total), and none were severe— Grade 3. The intensity was not recorded for 33 of the events (4.1%) (Table 8). There were no differences between the different groups concerning the intensity of the events. For example, 93.0% of events were mild in the BFD group, 91.8% in the INN group, and 97.6% in the STD group.

Most of the gastrointestinal symptoms were mild, with one growth failure within the BFD group that led to the introduction of artificial breastfeeding together with breastfeeding; two events, neonatal constipation and abdominal pain, recorded for one infant within the INN group, led to a change in formula and the abandonment of the study; and two (infantile colic and infant reflux) were recorded for one infant within the STD group and led to a change in formula and the abandonment of the study (Table 8).

For morbidities that occurred with a frequency greater than 1%, BFD infants exhibited the lowest incidence and infants fed the INN formula experienced significantly fewer general disorders and disturbances compared with the STD group. Indeed, the infants who were fed STD formula had a significantly higher incidence of atopic dermatitis, bronchitis, and bronchiolitis events than the infants who were fed BFD or INN formula. (Table 9).

## 4. Discussion

The present randomized, double-blind, placebo-controlled clinical trial was designed to determine whether a novel starting infant formula with reduced protein content and lower casein to whey protein ratio by increasing the content of α-lactalbumin influenced weight gain and body composition compared to a standard formula at 6 and 12 months of age. Furthermore, this product contains higher levels of DHA and ARA, as well as a postbiotic thermally inactivated (BPL1^TM^ HT), compared with standard infant formula. An exclusively breastfed population was followed up as a reference. At 6 and 12 months, infants receiving either INN or STD formula gained more weight than the BFD group, while no difference was observed between STD and INN formulas. BMI was also higher in infants fed either formula than in those breastfed. Regarding body composition, length, head circumference, and tricipital/subscapular skinfolds, we found that all of the measures were similar between all groups. It is important to note that the INN formula was considered safe based on weight gain and body composition, which were within the normal limits, according to WHO standards. Compared to both formulas, BFD produced stools that were more liquid in consistency. Throughout the study period, all groups exhibited similar digestive tolerance and infant behavior. In terms of the total number of adverse events reported, the STD formula had the highest number, followed by the INN formula and finally, the BFD group. However, the majority of them were not related to the type of feeding. Besides, infants fed either BFD or the INN formula exhibited significantly lower episodes of atopic dermatitis, bronchitis, and bronchiolitis events than those fed the STD formula.

In addition to exposure to metabolic and endocrine factors during pregnancy [47], protein intake in formula-fed infants during the first year of life may have a significant impact on growth later in life and the risk of obesity and metabolic disorders in adulthood [20,48]. Indeed, the protein content of infant formula is usually greater than that of human milk to provide all essential amino acids in adequate quantities [49]. Scientific evidence indicates that infants fed a formula containing more protein gain more weight during the first year of life and are heavier at 2 years of age than infants consuming a formula containing less protein [48], which reduces the risk of obesity at school age [24]. In our RCT, infants fed either INN or STD formula gained more weight than the breastfed group at 6 months as measured by differential daily weight gain per day (g). Similar results have been reported in other earlier studies [50] even with a relatively low content of protein [51]. However, previous studies on formulas with a reduced protein-to-energy ratio of up to 1.8 g/100 kcal have shown more modest differences in growth patterns compared with breastfed infants, similar to our findings [48,52,53], which supports the hypothesis that the protein-to-energy ratio plays a key role for weight gain. In our study the intake of INN formula was slightly higher than the STD formula, thus compensating for the lower energy content and protein of the former. Compensation for lower energy density and protein has been reported in other studies [51]. After 12 months, the STD group was significantly higher than the BFD and INN groups. We should note that the INN formula contains 8% less protein per 100 kcal than the STD. Thus, in the first six to eight weeks of life, there is almost no difference between breast milk and formula-fed infants in terms of growth (gain in weight and length). Indeed, it has been previously reported that formula-fed infants gained weight and length more rapidly than breastfed infants from about two months of age to the end of their first year of life [54]. Interestingly, the results of a recent review suggest that the difference in weight gain between formula-fed and breastfed infants is relatively small, comparable, and significantly less than the nutritionally significant differences used in determining sample size [55].

In addition to the total protein content, the whey/casein ratio in infant formulas seems to be important for the first year of life. Whey proteins, particularly β-lactoglobulin and α-lactalbumin, are rapidly digested and participate in the building of muscle mass [25], and particularly, α-lactalbumin is the major protein in human milk [26]. Thus, the addition of bovine α-lactalbumin to infant formula modifies the plasma amino acid pattern of the receiver infant, allowing a reduction in the protein content of the formula. On the other hand, casein proteins are water-insoluble high-molecular-weight molecules that are a source of phosphate and calcium in human milk, because of the highly phosphorylated nature of β-casein and α-S1-casein, and the requirement for calcium in forming the aggregates of casein micelles [56]. Nevertheless, the amount should be controlled to avoid improper digestion in early life. In this context, using cow’s milk as the protein source in infant formula might result in an α-casein-dominant formula, which differs from whey-protein dominance in human milk. Further, bovine whey contains a high concentration of β-lactoglobulin, which is absent in human milk [57]; therefore, adding bovine α-lactalbumin to infant formula makes it more similar to that of breastfed infants [58]. Moreover, α-lactalbumin is relatively rich in tryptophan, which may result in satisfactory growth and plasma tryptophan levels similar to those of breastfed infants and infants fed standard formula [59]. Nevertheless, in our study, both formula-fed infants gained weight more rapidly compared with exclusively breastfed infants, and there was no significant difference between the groups in terms of length, head circumference, tricipital and subscapular skinfolds, and upper arm circumference, suggesting that both formulas cause similar body composition in infants.

The proportion of fatty acids present in infant formulas is important to ensure the correct growth and development of infants. In the last 20 years, formulas have been supplemented with LC-PUFA in amounts similar to breast milk. Despite the new EU regulation that indicates that AA does not need to be included, intervention studies assessing the impact of DHA- and AA-supplemented formulas have resulted in numerous positive developmental outcomes (closer to breastfed infants) including measures of specific cognition functions, visual acuity, and immune responses [60,61,62]. AA has different biological functions compared to DHA, for example, AA has exclusive functions in the vasculature and specific aspects of immunity. Undeniably, most of the trials include both DHA and AA, and test development specific to DHA such as neural and visual development. DHA suppresses membrane AA concentrations and its function. Infant formula with DHA and no AA runs the risk of cardio- and cerebrovascular morbidity and even mortality through suppression of the favorable oxylipin derivatives of AA [63]. International expert consensus suggests that infant formulas that contain DHA at a concentration of 0.3% to 0.5% by weight of total fat and AA at a minimum level equivalent to the amount of DHA should be clinically tested [29,30]. In this study, we used an advanced formula enhanced with a double amount of DHA/AA in comparison with an STD formula to provide evidence about weight gain and body composition, among other outcomes. Throughout the study, the percentage of children with high tolerance increased without significant differences between formulations and compared to the group receiving BFD. According to the guardians or parents, the infant’s behavior was classified as altered mood or pleasant mood. There was a decrease in the number of infants with altered behavior throughout the study. The three methods of feeding did not show any significant differences.

The INN formula was also supplemented with a thermally inactivated postbiotic *Bifidobacterium animalis* subsp. *lactis*, BPL1^TM^ [40,64], which may confer some benefits regarding body composition, metabolism, and gut microbiota composition. Probiotics have many health benefits by modulating the gut microbiome; nonetheless, techno-functional limitations have made it gradually shift from viable probiotic bacteria towards non-viable postbiotics, paraprobiotics, and/or probiotics-derived biomolecules, so-called postbiotics [64]. Since the experimental formula of the present study was supplemented with BPL1^TM^ HT, it should be noted that in vivo studies reported that this strain can be considered a Generally Recognized as Safe (GRAS) substance. In terms of the intensity of GI symptoms, tolerability, and safety, there were no differences between the different groups (BFD, STD, and INN formula). In different contexts and using different methods, infants are fed a variety of products. The formula for infants is available in many variations, many of which are superior to other methods of feeding and some of which are inferior to others. On the other hand, there was a significant difference between BFD group presenting stools of a more liquid consistency, although this difference no longer existed at visit five (12 months). Additionally, BFD infants had more stools per day at visits 1, 2, and 3, but these differences ceased to be significant at visits 4 and 5. STD formula group reported the most GI symptoms, followed by the INN formula and BFD group. Triacylglycerol composition of infant formulas can influence the characteristics of stools i.e., appearance and consistency. As the fat sources differed between the STD and the INN formulas, we can hypothesize that the changes observed in fecal consistency might be due in part to this feature.

There were no differences in safety across the study between the different groups. Several differences were observed between the two formulas, showing a significant reduction in general disorders and disturbances among children who received the INN formula. STD formula-fed infants were more likely to cause atopic dermatitis, bronchitis, and bronchiolitis than BFD or INN formula-fed infants. Eczema or atopic dermatitis is a common chronic inflammatory skin disease, mostly occurring in children. Indeed, a meta-analysis showed that probiotic supplementation during both the prenatal and the postnatal period reduced the incidence of atopic dermatitis in infants and children, suggesting that starting probiotic treatment during gestation and continuing through the first 6 months of the infant’s life may be of benefit in the prevention of atopic dermatitis [65]. In this context, a study demonstrated that prenatal and postnatal supplementation with a mixture of *B. bifidum* BGN4, *B. lactis* AD011, and *L. acidophilus* AD031 is an effective approach in preventing the development of eczema in infants at high risk of allergy during the first year of life [66]. Hence, the lower incidence of atopic dermatitis observed in the INN formula group compared to the STD group might be mediated by the supplementation of the postbiotic BPL1^TM^ HT. Likewise, respiratory tract infections represent one of the main health problems in children of different ages [67]. Finally, changes in the intestinal microbiota will need to be assessed.

There are some limitations to this study. There were no significant correlations between the type of feeding and the number of adverse events. Nevertheless, the number of infants in the present study was calculated for growth as the main variable and not for morbidity.

In conclusion, this clinical trial involved the evaluation of a novel infant formula with reduced content of total protein and modification of the whey/casein ratio by increasing the content of α-lactalbumin, increased levels of both AA and DHA, and postbiotic in comparison with standard infant formula. For exploratory analysis, a third unblinded group of breastfed infants was used. Both formulas gained more weight at 6 and 12 months than the BFD group, while no differences were observed between STD and INN formulas. Infants fed both formulas had a higher BMI than those fed BFD. Body composition, head circumference, and tricipital/subscapular skinfolds were similar between the two groups. We should note that the INN formula is safe and it showed a reduction in atopic dermatitis, bronchitis, and bronchiolitis in infants compared to STD formula. The consistency of the stools produced by BFD was more liquid in comparison with both formulas. For further analysis, it would be necessary to examine more precise health biomarkers and to carry out long-term longitudinal studies.

## Figures and Tables

**Figure 1 nutrients-15-00147-f001:**
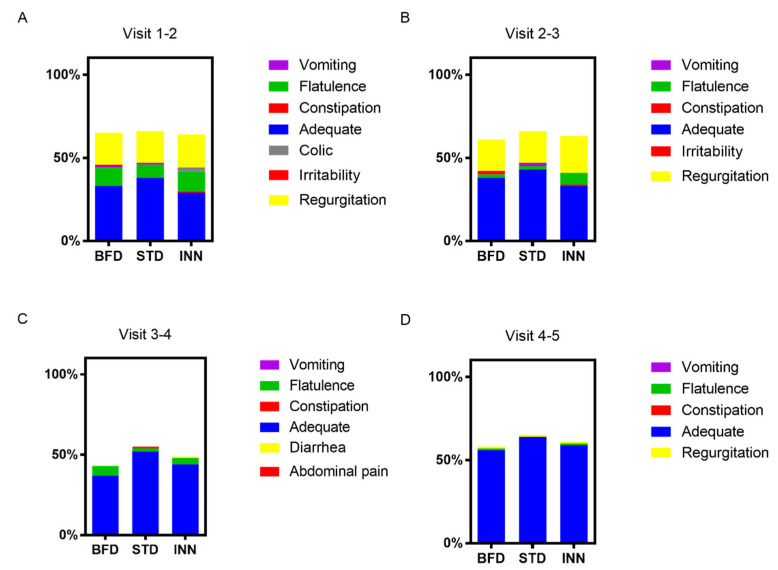
Digestive tolerance. Data are presented as percentages (%). BFD, breastfeeding group; STD, standard group; INN, INNOVA group. (**A**). The digestive events that occurred during visits 1 and 2, (**B**). The digestive events that occurred during visits 2 and 3, (**C**). The digestive events that occurred during visits 3 and 4, (**D**). The digestive events that occurred during visits 4 and 5.

**Table 1 nutrients-15-00147-t001:** Nutritional composition of the standard infant formula (STD) and study formula (INN).

Composition	STD Formula			INN Formula		
	100 g	100 mL	100 kcal	100 g	100 mL	100 kcal
Energy (kcal)	514	67		489	67	
Energy (kJ)	2152	280		2046	279	
Total Fat (g)	27	3.5	5.2	25.6	3.5	5.2
Linoleic acid (ꞷ-6) (mg)	3.79	493	736	3.147	427	641
α-linolenic acid (ꞷ-3) (mg)	500	65	97	387	53	79
Arachidonic acid AA (ꞷ-6) (mg)	53	6.9	10	118	16	24
Docosahexaenoic acid DHA (ꞷ-3) (mg)	53	6.9	10	118	16	24
Carbohydrates (g)	56.4	7.3	11	54.6	7.4	11.1
Total sugars (g)	56.4	7.3	11	53	7.2	10.8
Lactose (g)	55	7.2	10.7	53	7.2	10.8
Galacto-oligosaccharides (g)	3.1	0.4	0.6	2.9	0.4	0.6
Proteins (g)	10.6	1.4	2.1	9.4	1.3	1.9
Whey proteins (g)	6.4	0.83	1.2	6.6	0.9	1.3
Caseins (g)	4.2	0.55	0.82	2.8	0.38	0.57

Fat source: STD formula, a mixture of vegetable oils (palm olein, coconut oil, canola oil, high-oleic sunflower oil, sunflower oil, and palm oil); INN formula, a blend of milk lipids, a mixture of vegetable oils (sunflower oil, canola oil, and coconut oil), and purified fish oil.

**Table 2 nutrients-15-00147-t002:** Baseline demographics of the infants in the study.

	BFD	STD	INN	*p*-Value
N	70	70	70	
Gender (% of women)	28 (40.0)	29 (41.4)	33 (47.1)	0.738
Gestational age (weeks)	39.44 (1.1)	39.20 (1.1)	39.34 (0.9)	0.389
Birth weight (g)	3359.20 (403.9)	3300.87 (462.9)	3363.30 (408.7)	0.626
*Birth weight > 3500 g (%)*	22 (31.4)	22 (31.4)	27 (38.6)	0.613
Childbirth delivery *n*, (%)				0.579
*Cesarean section*	13 (18.6)	17 (24.3)	20 (28.6)	
*Dystocic*	8 (11.4)	6 (8.6)	9 (12.9)	
*Normal childbirth*	49 (70.0)	47 (67.1)	41 (58.6)	
Assisted reproductive technology *n*, (%)	4 (5.7)	6 (8.6)	3 (4.3)	0.679
*Assisted reproductive technology type (%)*				0.066
*In vitro fertilization*	1 (25.0)	6 (100.0)	2 (66.7)	
*Artificial insemination*	2 (50.0)	0 (0.0)	1 (33.3)	
*Egg donation*	1 (25.0)	0 (0.0)	0 (0.0)	
Maternal BMI (kg/m^2^), (x, SD)	23.72 (3.2)	23.92 (3.3)	23.78 (3.0)	0.967
*BMI mother>25 n, (%)*	22 (31.4)	22 (31.4)	24 (34.3)	0.85
Mother’s medical background *n*, (%)				0.849
*Allergy*	0 (0.0)	0 (0.0)	2 (2.9)	
*Hypothyroidism*	7 (10.0)	5 (7.1)	4 (5.7)	
*Arterial hypertension*	0 (0.0)	2 (2.9)	1 (1.4)	
*No*	58 (82.9)	55 (78.6)	55 (78.6)	
*Others*	5 (7.1)	8 (11.4)	8 (11.4)	
Visit 1, (21 days)	BFD	STD	INN	*p*-value
Depositions per day	5.2 (3.2)	2.5 (2.0)	2.3 (1.4)	<0.001 *
Weight	3921.8 (475.5)	3805.0 (507.5)	3916.6 (453.9)	0.268
Length	53.1 (1.9)	52.6 (2.2)	52.8 (1.7)	0.287
Head circumference	36.6 (1.0)	36.2 (1.3)	36.5 (1.2)	0.214

Data are expressed as mean and standard deviation, and frequency or percentage. BMI, body mass index; NS, non-significant. * *p* < 0.05, statistical differences among groups.

**Table 3 nutrients-15-00147-t003:** Differential daily weight gain and absolute weight in the entire study.

**Differential Daily Weight Gain per Day (g)**						
**Groups**	**BFD**	**STD**	**INN**	***p*-Value**		
**Differential daily weight gain**	**(*n* = 58)**	**(*n* = 65)**	**(*n* = 62)**	**STD vs. BFD**	**INN vs. BFD**	**STD vs. INN**
Visit 4 vs Visit 1	22.0 ± 4.6	24.24 ± 5.06	24.60 ± 5.29	0.01	0.004	0.693
Visit 5 vs visit 1	16.01 ± 2.66	17.41 ± 3.25	17.00 ± 2.95	0.011	0.057	0.457
**Weight (g)**						
**Groups**	**BFD**	**STD**	**INN**	***p*-Value**		
**Weight**	**(*n* = 58)**	**(*n* = 65)**	**(*n* = 62)**	**Formula**	**Visit**	**Formula x visit**
Visit 1, 21 days	3951.3 ± 498.0	3805.8 ± 509.6	3916.9 ± 468.7	0.081	0.848	0.011
Visit 2, 2 months	5285.1 ± 564.9	5193.7 ± 684.0	5236.6 ± 623.2			
Visit 3, 4 months	6558.6 ± 697.1	6733.0 ± 937.7	6843.1 ± 855.1			
Visit 4, 6 months	7471.1 ± 846.3	7718.7 ± 1132.8	7903.1 ± 1063.5			
Visit 5, 12 months	9505.2 ± 985.6	9869.7 ± 1382.1	9797.6 ± 1234.4			

Data are expressed as mean ± standard deviation.

**Table 4 nutrients-15-00147-t004:** Length and head circumference in the entire study.

**Length (cm)**			
**Visits**	**BFD**	**STD**	**INN**
	**(*n* = 58)**	**(*n* = 65)**	**(*n* = 62)**
Visit 1	53.15 ± 2.08	52.49 ± 2.24	52.87 ± 1.78
Visit 2	57.89 ± 2.42	57.47 ± 2.33	57.95 ± 2.02
Visit 3	63.16 ± 2.42	63.24 ± 3.67	63.50 ± 2.16
Visit 4	66.68 ± 2.64	67.01 ± 3.26	67.15 ± 2.14
Visit 5	75.44 ± 2.90	75.82 ± 3.08	75.96 ± 2.54
**Head circumference (cm)**			
**Visits**	**BFD**	**STD**	**INN**
	**(*n* = 58)**	**(*n* = 65)**	**(*n* = 62)**
Visit 1	36.66 ± 1.02	36.22 ± 1.34	36.51 ± 1.25
Visit 2	36.09 ± 1.17	38.87 ± 1.36	39.11 ± 1.63
Visit 3	41.56 ± 1.12	41.79 ± 3.73	41.64 ± 1.39
Visit 4	43.19 ± 1.22	43.18 ± 1.37	43.41 ± 1.45
Visit 5	46.40 ± 1.18	46.25 ± 1.53	46.29 ± 1.50

Data are expressed as mean and standard deviation.

**Table 5 nutrients-15-00147-t005:** Tricipital and subscapular skinfolds and mean upper arm circumference in visits 3, 4, and 5.

**Tricipital Skinfold (mm)**			
	**BFD**	**STD**	**INN**
	**(*n* = 58)**	**(*n* = 65)**	**(*n* = 62)**
Visit 3	12.32 ± 2.26	12.66 ± 2.67	12.78 ± 3.00
Visit 4	12.35 ± 2.23	12.60 ± 2.65	12.90 ± 2.99
Visit 5	12.34 ± 2.23	12.63 ± 2.65	12.84 ± 2.99
**Subscapular skinfold (mm)**			
	**BFD**	**STD**	**INN**
	**(*n* = 58)**	**(*n* = 65)**	**(*n* = 62)**
Visit 3	9.92 ± 2.69	9.83 ± 2.72	10.25 ± 2.73
Visit 4	9.90 ± 2.49	9.94 ± 2.72	10.36 ± 2.73
Visit 5	9.91 ± 2.58	9.89 ± 2.73	10.30 ± 2.72
**Mean upper arm circumference (cm)**			
	**BFD**	**STD**	**INN**
	**(*n* = 58)**	**(*n* = 65)**	**(*n* = 62)**
Visit 3	13.79 ± 1.30	13.89 ± 1.67	14.15 ± 1.21
Visit 4	14.41 ± 1.24	14.72 ± 1.46	14.98 ± 1.88
Visit 5	15.34 ± 1.26	15.29 ± 1.68	15.94 ± 1.39

Data are expressed as mean and standard deviation.

**Table 6 nutrients-15-00147-t006:** Body mass index, body fat percentage, and lean mass.

**Body Mass Index (kg/m^2^)**				***p*-Value**		
**Visits**	**BFD**	**STD**	**INN**	**Formula**	**Visit**	**Formula x Visit**
	**(*n* = 58)**	**(*n* = 65)**	**(*n* = 62)**			
Visit 1	13.94 ± 1.11	13.75 ± 1.04	13.97 ± 1.07	0.487	<0.001	<0.001
Visit 2	15.74 ± 1.08	15.67 ± 1.30	15.56 ± 1.28			
Visit 3	16.42 ± 1.22	16.97 ± 3.61	16.94 ± 1.59			
Visit 4	16.78 ± 1.30	17.15 ± 1.83	17.49 ± 1.81			
Visit 5	16.69 ± 1.28	17.11 ± 1.64	16.94 ± 1.54			
**Body fat percentage**				***p*-value**		
**Visits**	**BFD**	**STD**	**INN**	**Formula**	**Visit**	**Formula x visit**
	**(*n* = 58)**	**(*n* = 65)**	**(*n* = 62)**			
Visit 3	19.35 ± 3.53	19.60 ± 3.39	20.37 ± 3.67	0.249	0.005	0.958
Visit 4	20.36 ± 3.53	20.30 ± 3.73	21.29 ± 3.93			
Visit 5	20.84 ± 3.64	21.02 ± 3.82	21.61 ± 4.01			
**Lean mass (g)**				***p*-value**		
**Visits**	**BFD**	**STD**	**INN**	**Formula**	**Visit**	**Formula x visit**
	**(*n* = 58)**	**(*n* = 65)**	**(*n* = 62)**			
Visit 3	5284 ± 559	5403 ± 709	5436 ± 615	0.215	<0.001	0.054
Visit 4	5942 ± 654	6135 ± 823	6207 ± 782			
Visit 5	7509 ± 703	7805 ± 1023	7670 ± 967			

Data are expressed as mean and standard deviation.

**Table 7 nutrients-15-00147-t007:** The daily number of stools.

Number of Daily Depositions	Visit 1	Visit 2	Visit 3	Visit 4	Visit 5
BFD					
Mean (SD)	5.2 (3.2)	3.1 (2.3)	2.3 (1.5)	2.0 (1.4)	1.8 (1.0)
Median [Min, Max]	5 [3, 8]	3 [1, 4]	2 [1, 3]	2 [1, 2]	2 [1, 2]
STD					
Mean (SD)	2.5 (2.0)	1.5 (0.7)	1.3 (0.6)	1.5 (1.0)	1.9 (0.9)
Median [Min, Max]	2 [1, 4]	1 [1, 2]	1 [1, 2]	1 [1, 2]	2 [1, 2]
INN					
Mean (SD)	2.3 (1.4)	1.6 (0.8)	1.6 (0.9)	1.6 (0.9)	1.9 (0.8)
Median [Min, Max]	2 [1, 3]	1 [1, 2]	1 [1, 2]	1 [1, 2]	2 [1, 2]

Data are expressed as mean, maximum, and minimum.

**Table 8 nutrients-15-00147-t008:** Gastrointestinal symptoms across the study.

**Total adverse events**	**BFD (*n* = 70)**	**STD (*n* = 70)**	**INN (*n* = 70)**
Subjects with at least one adverse event (%)	62 (88.6%)	57 (81.4%)	61 (87.1%)
Total number	227	291	282
Mild—Grade 1	211	284	259
Moderate—Grade 2	3	3	7
Severe—Grade 3	0	0	0
Potentially fatal—Grade 4	0	0	0
Lethal—Grade 5	0	0	0
Unknown	13	4	16
**Adverse events related to feeding**	**BFD (*n* = 70)**	**STD (*n* = 70)**	**INN (*n* = 70)**
Subjects with at least one adverse event (%)	1 (1.4%)	1 (1.4%)	1 (1.4%)
Total number	1	2	2
Mild—Grade 1	1	2	2
Moderate—Grade 2	0	0	0
Severe—Grade 3	0	0	0
Potentially fatal—Grade 4	0	0	0
Lethal—Grade 5	0	0	0
Unknown	0	0	0

Data are expressed as counts and percentages.

**Table 9 nutrients-15-00147-t009:** Morbidities that occurred with a frequency greater than 1%.

MedDRA Dictionary Version 21.0 (MedDRA, 2018)	BFD (*n* = 70)		STD (*n* = 70)		INN (*n* = 70)	
	Cases	%	Cases	%	Cases	%
Cold	42	18.50	47	16.15	38	13.48
Pyrexia	16	7.05	24	8.25	11	3.90
Bronchitis/Bronchiolitis*	13	5.7	34	11.7	17	6.00
Upper respiratory tract infection	11	4.85	11	3.78	15	5.32
Infantile colic	14	6.17	9	3.09	10	3.55
Conjunctivitis	5	2.20	11	3.78	15	5.32
Neonatal constipation	6	2.64	10	3.44	9	3.19
Acute otitis media	4	1.76	6	2.06	13	4.61
Rhinorrhea	9	3.96	5	1.72	6	2.13
Gastroenteritis	4	1.76	8	2.75	8	2.84
Cough	9	3.96	7	2.41	5	1.77
Vaccination complication	1	0.44	10	3.44	7	2.48
Laryngitis	1	0.44	9	3.09	7	2.48
Atopic dermatitis *	1	0.44	11	3.78	2	0.71
Seborrheic dermatitis	2	0.88	4	1.37	5	1.77
Neonatal diarrhea	2	0.88	3	1.03	5	1.77
Oral candidiasis	4	1.76	3	1.03	2	0.71
Nasopharyngitis	1	0.44	2	0.69	6	2.13
Infantile eczema	1	0.44	5	1.72	2	0.71
Diaper dermatitis	2	0.88	2	0.69	4	1.42
Rash	3	1.32	2	0.69	2	0.71
Neonatal disorder	2	0.88	1	0.34	4	1.42
Upper respiratory tract infection	2	0.88	1	0.34	3	1.06
Wheezing	2	0.88	3	1.03	1	0.35
Infant reflux	1	0.44	2	0.69	3	1.06
Otitis media	4	1.76	1	0.34	1	0.35
Gastroesophageal reflux disease	4	1.76	0	0.00	1	0.35
Heart murmur	4	1.76	1	0.34	0	0.00
Diarrhea	1	0.44	0	0.00	4	1.42
Atopic Dermatitis*	1	0.44	3	1.03	0	0.00
Traumatic head injury	0	0.00	1	0.34	4	1.42
Abdominal pain	0	0.00	1	0.34	3	1.06
Bronchial hyperreactivity	0	0.00	3	1.03	0	0.00
Decreased appetite	0	0.00	3	1.03	0	0.00
Pharyngotonsillitis	0	0.00	1	0.34	3	1.06
Umbilical hernia	0	0.00	1	0.34	3	1.06
Viral rash	0	0.00	1	0.34	3	1.06
Traumatic injury to the mouth	1	0.44	0	0.00	3	1.06
Total	173		246		225	

Data are expressed as counts and percentages, * *p* < 0.05 BFD and INN vs. STD group. For the analysis, the Chi-square test was used with the Yates correction where appropriate. Bronchitis/bronchiolitis showed a p-value of 0.029 (BFD vs. STD), 1.0 (BFD vs. INN), and 0.026 (STD vs. INN). Atopic dermatitis exhibited a *p*-value of 0.027 (BFD vs. STD), 1.0 (BFD vs. INN), and 0.029 (STD vs. INN).

## Data Availability

The datasets used and/or analyzed during the current study are available from the corresponding author upon reasonable request.

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
