# Peer review of "Effects of a Novel Infant Formula on Weight Gain, Body Composition, Safety and Tolerability to Infants: The INNOVA 2020 Study"

_nutrients, 2022, doi:10.3390/nu15010147_

Round 1

Reviewer 1 Report

The present study entitled “Effects of a Novel Infant Formula on Weight Gain, Body Composition, Safety and Tolerability of Infants: The INNOVA 2020 Study” aimed to evaluate a novel starting formula on weight gain and body composition of infants up to 6 and 12 as well as safety and tolerability, intervention: involved the evaluation of a novel infant formula with reduced content of total protein and modification of the whey/casein ratio by increasing the content of α-lactalbumin, increased levels of both AA and DHA, and postbiotic in comparison with standard infant formula. The authors compared the standard formula vs a novel formula and uses breastfed infants as a reference group. A detailed comparison of the incidence of a number of diseases was made and the frequency of common gastrointestinal symptoms was also reported. The methods used, results, and data interpretation look good. I have the following minor comments:

1-Why did you specifically use alpha-lactoalbumin?

2- The abstract is very long and should be shortened.

3- Line 501: Please, correct the following: “similar ot our findings…”

4-Line 218: Please, correct the following: “not being breasfed…”

Author Response

Dear Ms. Katarzyna Kieć

Assistant Editor,

Thank you for providing us with the opportunity to submit a revised version of our manuscript entitled Effects of a Novel Infant Formula on Weight Gain, Body Composition, Safety and Tolerability of Infants: The INNOVA 2020 Studyto Nutrients.

The authors thank the reviewers for their thoughtful comments and suggestions on our manuscript. We have considered all of the comments and incorporated them into the revised manuscript. Changes to the original document are highlighted as tracked changes, and an itemized point-by-point response to the comments from reviewers is presented below.

COMMENTS FROM REVIEWER #1

Comment #1

The present study entitled “Effects of a Novel Infant Formula on Weight Gain, Body Composition, Safety and Tolerability of Infants: The INNOVA 2020 Study” aimed to evaluate a novel starting formula on weight gain and body composition of infants up to 6 and 12 as well as safety and tolerability, intervention: involved the evaluation of a novel infant formula with reduced content of total protein and modification of the whey/casein ratio by increasing the content of α-lactalbumin, increased levels of both AA and DHA, and postbiotic in comparison with standard infant formula. The authors compared the standard formula vs a novel formula and uses breastfed infants as a reference group. A detailed comparison of the incidence of a number of diseases was made and the frequency of common gastrointestinal symptoms was also reported. The methods used, results, and data interpretation look good. I have the following minor comments. Why did you specifically use alpha-lactoalbumin?

Response: Thanks to the reviewer for his/her kind comment on our manuscript. By adding alpha-lactalbumin, the major protein in breast milk, the overall protein content of infant formula can be reduced mimicking human milk in terms of protein concentration and composition (PMID: 21063429). This aspect has been addressed in the discussion (page 16, lines 617-638) and states “In addition to the total protein content, the whey/casein ratio in infant formulas seems to be important for the first year of life. Whey proteins, particularly β-lactoglobulin and α-lactalbumin, are rapidly digested and participate in the building of muscle mass [25], and particularly, α-lactalbumin is the major protein in human milk [26]. Thus, the addition of bovine α-lactalbumin to infant formula modifies the plasma amino acid pattern of the receiver infant, allowing a reduction in the protein content of the formula. On the other hand, casein proteins are water-insoluble high molecular weight molecules that are a source of phosphate and calcium in human milk, because of the highly phosphorylated nature of β-casein and α-S1-casein, and the requirement for calcium in forming the aggregates of casein micelles [56]. Nevertheless, the amount should be controlled to avoid improper digestion in early life. In this context, using cow’s milk as the protein source in infant formula might result in an α-casein-dominant formula, which differs from the whey protein dominance in human milk. Further, bovine whey contains a high concentration of β-lactoglobulin, which is absent in human milk [57], therefore, adding bovine α-lactalbumin to infant formula makes it more similar to that of breastfed infants [58]. Moreover, α-lactalbumin is relatively rich in tryptophan, which may result in satisfactory growth and plasma tryptophan levels similar to those of breastfed infants and infants fed standard formula [59]. Nevertheless, in our study, both formula-fed infants gained weight more rapidly compared with exclusively breastfed infants, and there was no significant difference between the groups in terms of length, head circumference, tricipital and subscapular skinfolds, and upper arm circumference, suggesting that both formulas cause similar body composition in infants.

Comment #2

The abstract is very long and should be shortened.

Response: Using the reviewer's comment, the abstract was shortened and now states (page 1-2 lines 37-133), “Exclusive breastfeeding is recommended for the first six months of life to promote adequate infant growth and development, and to reduce infant morbidity and mortality, However, whenever some mothers are not able to breastfeed their infants, infant formulas mimicking human milk are needed, and the safety and efficacy of each formula should be tested. Here, we report the results of a multicenter, randomized, blinded, controlled clinical trial that aimed to evaluate a novel starting formula on weight gain and body composition of infants up to 6 and 12 months, as well as safety and tolerability. For the intervention period, infants were divided into three groups: group 1 received the formula 1 (Nutribén® Innova1 or INN (n=70)), with a lower amount of protein, and enriched in α-lactalbumin protein, and with a double amount of docosahexaenoic   acid/arachidonic acid than the standard formula; it also contained a thermally inactivated postbiotic (Bifidobacterium animalis subsp. lactis, BPL1TM HT). Group 2 received the standard formula or formula 2 (Nutriben® Natal or STD (n=70)) and the third group was exclusively breastfed for exploratory analysis and used as a reference (BFD group (n=70)). During the study, visits were made at 21 days, 2, 4, 6, and 12 months of age. Weight gain was higher in both formula groups than in the BFD group at 6 and 12 months, whereas no differences were found between STD and INN groups neither at 6 nor at 12 months. Likewise, body mass index was higher in infants fed the two formulas compared with the BFD group. Regarding body composition, length, head circumference and tricipital/subscapular skinfolds were alike between groups. The INN formula was considered safe as weight gain and body composition were within the normal limits, according to WHO standards. The BFD group exhibited more liquid consistency in the stools   compared to both formula groups. All groups showed similar digestive tolerance and infant behavior. However, a higher frequency of gastrointestinal symptoms was reported by the STD formula group (n=291), followed by the INN formula (n=282) and the BFD groups (n=227). There were fewer respiratory, thoracic, and mediastinal disorders among BFD children. Additionally, infants receiving the INN formula experienced significantly fewer general disorders and disturbances than those receiving the STD formula. Indeed, atopic dermatitis, bronchitis, and bronchiolitis were significantly more prevalent among infants who were fed the STD formula compared to those fed INN formula or breastfed. To evaluate whether there are significant differences between formula treatments, beyond growth parameters, it would seem necessary to examine more precise health biomarkers and to carry out long-term longitudinal studies.

Comment #3

Line 501: Please, correct the following: “similar ot our findings…”

Response: Using the reviewer’s comment, the sentence was modified.

Comment #4

Line 218: Please, correct the following: “not being breasfed…”

Response: Using the reviewer’s comment, the sentence was modified.

Reviewer 2 Report

This manuscript sets out to examine the effects of a Novel Infant Formula on Weight Gain, Body Composition, Safety and Tolerability of Infants ' The topic of this paper is interesting. However, I feel a number of key aspects are missing in this study and this paper needs after minor revision before consideration for publication in the journal nutrients.

 1.Please refine the Abstract

 2.Please introduce what is the novelty of Novel Infant Formula in Abstract

 3.More information is needed about the Nutritional composition of human milk.  

 4.Fat source of infant formula? bovine milk fat or vegetable fat? Fat is an important energy source for infants, and it is very important for their growth and development.

 5.As far as I know, the characteristics of feces are closely related to the triglyceride composition of infant formula. The author did not provide information on the triglyceride composition of two kinds of infant formulas, whether OPO structural ester was added?

Author Response

Dear Ms. Katarzyna Kieć

Assistant Editor,

Thank you for providing us with the opportunity to submit a revised version of our manuscript entitled Effects of a Novel Infant Formula on Weight Gain, Body Composition, Safety and Tolerability of Infants: The INNOVA 2020 Studyto Nutrients.

The authors thank the reviewers for their thoughtful comments and suggestions on our manuscript. We have considered all of the comments and incorporated them into the revised manuscript. Changes to the original document are highlighted as tracked changes, and an itemized point-by-point response to the comments from reviewers is presented below.

COMMENTS FROM REVIEWER #2

Comment #1

This manuscript sets out to examine the effects of a Novel Infant Formula on Weight Gain, Body Composition, Safety and Tolerability of Infants ' The topic of this paper is interesting. However, I feel a number of key aspects are missing in this study and this paper needs after minor revision before consideration for publication in the journal nutrients. Please refine the Abstract

Response: Thanks to the reviewer for his/her kind comment on our manuscript. (page 1-2 lines 37-133), “Exclusive breastfeeding is recommended for the first six months of life to promote adequate infant growth and development, and to reduce infant morbidity and mortality, However, whenever some mothers are not able to breastfeed their infants, infant formulas mimicking human milk are needed, and the safety and efficacy of each formula should be tested. Here, we report the results of a multicenter, randomized, blinded, controlled clinical trial that aimed to evaluate a novel starting formula on weight gain and body composition of infants up to 6 and 12 months, as well as safety and tolerability. For the intervention period, infants were divided into three groups: group 1 received the formula 1 (Nutribén® Innova1 or INN (n=70)), with a lower amount of protein, and enriched in α-lactalbumin protein, and with a double amount of docosahexaenoic   acid/arachidonic acid than the standard formula; it also contained a thermally inactivated postbiotic (Bifidobacterium animalis subsp. lactis, BPL1TM HT). Group 2 received the standard formula or formula 2 (Nutriben® Natal or STD (n=70)) and the third group was exclusively breastfed for exploratory analysis and used as a reference (BFD group (n=70)). During the study, visits were made at 21 days, 2, 4, 6, and 12 months of age. Weight gain was higher in both formula groups than in the BFD group at 6 and 12 months, whereas no differences were found between STD and INN groups neither at 6 nor at 12 months. Likewise, body mass index was higher in infants fed the two formulas compared with the BFD group. Regarding body composition, length, head circumference and tricipital/subscapular skinfolds were alike between groups. The INN formula was considered safe as weight gain and body composition were within the normal limits, according to WHO standards. The BFD group exhibited more liquid consistency in the stools   compared to both formula groups. All groups showed similar digestive tolerance and infant behavior. However, a higher frequency of gastrointestinal symptoms was reported by the STD formula group (n=291), followed by the INN formula (n=282) and the BFD groups (n=227). There were fewer respiratory, thoracic, and mediastinal disorders among BFD children. Additionally, infants receiving the INN formula experienced significantly fewer general disorders and disturbances than those receiving the STD formula. Indeed, atopic dermatitis, bronchitis, and bronchiolitis were significantly more prevalent among infants who were fed the STD formula compared to those fed INN formula or breastfed. To evaluate whether there are significant differences between formula treatments, beyond growth parameters, it would seem necessary to examine more precise health biomarkers and to carry out long-term longitudinal studies.

Comment #2

Please introduce what is the novelty of Novel Infant Formula in Abstract

Response: Thanks to the reviewer for his/her comment, the abstract was modified and now states (page 1, lines 44-47), “group 1 received the formula 1 (Nutribén® Innova1 or INN (n=70)), with a lower amount of protein, and enriched in α-lactalbumin protein, and with a double amount of docosahexaenoic acid/arachidonic acid than the standard formula; it also contained a thermally inactivated postbiotic (Bifidobacterium animalis subsp. lactis, BPL1TM HT).”

Comment #3

More information is needed about the Nutritional composition of human milk

Response: Using the reviewer’s comment the information was added and now states (page 2-3, lines 167-175), “In human milk, 87% of the mass consists of water, 1% protein, 4% lipids, and 7% carbohydrates (of which 1 to 2.4% are oligosaccharides). A variety of minerals are also present in it (calcium, phosphorus, magnesium, potassium, sodium, chloride, and several trace elements) as well as all necessary vitamins. Some minor proteins are more prevalent in human milk (lysozyme, lactoferrin, etc.) as well as the nonprotein nitrogen fraction (urea, free amino acids, including taurine). As a result, the protein content of human milk is low (10 g/L) [16].

Comment #4

Fat source of infant formula? bovine milk fat or vegetable fat? Fat is an important energy source for infants, and it is very important for their growth and development.

Response: Using the reviewer’s comment, the information was added in the footnote of Table 1 and now states (page 6, lines 332-334), “Fat source. STD formula, a mixture of vegetable oils (palm olein, coconut oil, canola oil, high-oleic sunflower oil, sunflower oil and palm oil); INN formula, a blend of milk lipids, a mixture of vegetable oils (sunflower oil, can-ola oil and coconut oil) and purified fish oil.

Comment #5

As far as I know, the characteristics of feces are closely related to the triglyceride composition of infant formula. The author did not provide information on the triglyceride composition of two kinds of infant formulas, whether OPO structural ester was added?

Response: The reviewer is right. Tryacilgycerol composition of infant formulas can influence the characteristics of stools i.e appearance and consistency. As the fat sources differed between the STD and the INN formulas, we can hypothesize that the changes observed in fecal consistency might be due in part to this feature. This has now been included in the discussion section and now states (page 17, Lines 681-685), “Triacylglycerol composition of infant formulas can influence the characteristics of stools i.e., appearance and consistency. As the fat sources differed between the STD and the INN formulas, we can hypothesize that the changes observed in fecal consistency might be due in part to this feature.”. Anyhow, we did not attempt to include OPO structural ester in any of the two study formulas.
